# Unsupervised classification of vertical profiles of dual polarization radar variables

Jussi Tiira[1] and Dmitri N. Moisseev[1,2]

[1]Institute for Atmospheric and Earth System Research / Physics, Faculty of Science, University of Helsinki, Helsinki, Finland
[2]Finnish Meteorological Institute, Helsinki, Finland

**Correspondence:** Jussi Tiira (jussi.tiira@helsinki.fi)

**Abstract.** Vertical profiles of polarimetric radar variables can be used to identify fingerprints of snow growth processes. In order to systematically study such manifestations of precipitation processes, we have developed an unsupervised classification method. The method is based on $k$-means clustering of vertical profiles of polarimetric radar variables, namely reflectivity, differential reflectivity and specific differential phase. For rain events, the classification is applied to radar profiles truncated at the melting layer top. For the snowfall cases, the surface air temperature is used as an additional input parameter. The proposed unsupervised classification was applied to 3.5 years of data collected by the Finnish Meteorological Institute Ikaalinen radar. The vertical profiles of radar variables were computed above the University of Helsinki Hyytiälä station, located 64 km east of the radar. Using these data, we show that the profiles of radar variables can be grouped into 10 and 16 classes for rainfall and snowfall events respectively. These classes seem to capture most important snow growth and ice cloud processes. Using this classification, main features of the precipitation formation processes, as observed in Finland, are presented.

## 1 Introduction

Globally, majority of precipitation both during winter and summer originates from ice clouds (Field and Heymsfield, 2015). At higher latitudes winter precipitation occurs in the form of snow, which can have a dramatic impact on human life (Juga et al., 2012). There are a number of challenges in remote sensing of winter precipitation or ice clouds, i.e. quantitative estimation of ice water content or precipitation rate (von Lerber et al., 2017), identification of dangerous weather conditions, etc. To address these challenges, advances in identifying and documenting the processes that take place in ice clouds are needed.

There are several pathways by which ice particles grow, such as vapor deposition, aggregation and riming. Occurrence of these processes depends on environmental conditions. Interpretation of radar observations is based on our understanding of the link between microphysical and scattering properties of hydrometeors. By identifying particle types in observations, we may conclude what processes took place. Currently, dual-polarization radar observations are used in fuzzy logic classification to identify dominant hydrometeor type present in a radar volume (e.g. Chandrasekar et al., 2013; Thompson et al., 2014). Such methods work very well for classification of hydrometeors of summer precipitations and some features of winter precipitation types. The main challenge is the lack of distinction in dual-polarization radar variables between some ice particle habits. For example, large low-density aggregates and graupel may have similar radar characteristics. Furthermore, these methods perform

classification on radar volume by volume basis, without taking into account surrounding observations. Recently, a modification for the hydrometeor classifiers was proposed to make the algorithms aware of the surrounding by incorporating measurements from neighbouring radar volumes (Bechini and Chandrasekar, 2015; Grazioli et al., 2015b). This step has greatly improved classification robustness, but aims to identify particle types instead of fingerprints of microphysical processes.

In the past 10 years, a number of studies reported signatures of ice growth processes in dual polarization radar observations. Kennedy and Rutledge (2011) have reported bands of increased values of specific differential phase, $K_{dp}$, and differential reflectivity, $Z_{DR}$ in Colorado snow storms. These bands took place at altitudes where ambient air temperature was around -15 °C and their occurrence was attributed to growth of dendritic crystals. Andrić et al. (2013) have implemented a simple steady state single column snow growth model to explain main features of the bands. It was also observed that the occurrence of $K_{dp}$ bands can be linked to heavier surface precipitation (Kennedy and Rutledge, 2011; Bechini et al., 2013). Moisseev et al. (2015) have advocated that the $K_{dp}$ bands occur only in precipitation systems with high enough cloud tops heights, where a large number of ice crystals can be generated either by heterogeneous or homogeneous ice nucleation. Using a larger dataset, Griffin et al. (2018) have shown that the $K_{dp}$ bands can be linked to formation of ice by homogeneous ice nucleation at cloud tops. Furthermore, it was shown that the $K_{dp}$ bands can be linked to onset of aggregation (Moisseev et al., 2015) which tend to occur more frequent in higher water vapor content environments (Schneebeli et al., 2013). In addition to the above-listed studies different aspects of these bands were presented by Trömel et al. (2014); Oue et al. (2018); Kumjian and Lombardo (2017). Besides $K_{dp}$ bands in the dendritic growth zone, several studies (e.g. Grazioli et al., 2015a; Sinclair et al., 2016; Kumjian et al., 2016; Giangrande et al., 2016) have reported $K_{dp}$ observations in the temperature region where Hallett-Mossop (H-M; Hallett and Mossop, 1974) rime splintering secondary ice production takes place (Field et al., 2016). Sinclair et al. (2016) have shown that such observations can be used to test representation of the secondary ice production in numerical weather prediction models. Other dual-polarization observations that show notable features are high $Z_{DR}$ regions surrounding the cores of snow generating cells (Kumjian et al., 2014) and at the top of ice clouds which can be linked to the presence of planar crystals and further to the presence of super-cooled liquid water providing very favourable conditions for their growth at these temperatures (Williams et al., 2015; Oue et al., 2016).

As presented above, the fingerprints of snow growth processes can occur in the form of bands in stratiform clouds, either embedded in the precipitation or on top of a cloud, or in the form of convective generating cells. To identify and document such features, a classification method that uses vertical profiles of dual-polarization radar observations can be used. In this study, we have developed such an unsupervised classification method based on $k$-means clustering of vertical profiles of polarimetric radar variables, namely reflectivity, differential reflectivity and specific differential phase. The proposed classification is applied to 3.5 years of data collected with the Finnish Meteorological Institute Ikaalinen radar.

The paper is structured as follows. Section 2 describes polarimetric radar and temperature data and their preprocessing. The unsupervised classification method is presented in Sect. 3. Section 4 is dedicated for the analysis and interpretation of the classification results and Sect. 5 presents the conclusions.

## 2 Data

In this study, we use vertical profiles of polarimetric radar observables of precipitation over Hyytiälä forestry station in Juu-
pajoki, Finland collected using Ikaalinen weather radar, hereafter IKA. The radar is located 64 km west from the station. The
measurements have been performed between January 2014 and May 2017, partly during the Biogenic Aerosols – Effects on
Clouds and Climate (BAECC; Petäjä et al., 2016) field campaign which took place at the measurement site in 2014.

The classification training material includes all precipitation events from this period, where, after preprocessing (see Sect. 2.2),
there were no major data quality problems identified. Since synoptic conditions may be similar even in cases where there are
gaps in observed precipitation, we define any two precipitation events separate if a continuous gap in reflectivity between them
exceeds 12 hours. See Sect. 4 for more discussion. During the observation period, we have identified 74 snow and 123 rain
events that meet these conditions. Generally, the full temporal extent of an event includes radar profiles in which precipitation
have not reached the ground. A list of the precipitation events is given in supplement S1.

In order to link features identified in vertical profiles of radar variables to precipitation processes, information on the ambient
temperature is needed. For this purpose we use vertical profiles of temperature from the National Center for Environmental
Prediction (NCEP) Global Data Assimilation System (GDAS) output for Hyytiälä interpolated to match the temporal and
vertical resolution of the vertical profiles of radar variables used in this study. The original temporal resolution of the NCEP
GDAS data over Hyytiälä is 3 hours, and the vertical resolution is 25 hPa between the 1000 and 900 hPa levels, and 50 hPa
elsewhere.

### 2.1 Vertical profiles of dual-polarization radar observables

The radar profiles are extracted from IKA C-band radar range height indicator (RHI) measurements. IKA performs RHI scans
directly towards Hyytiälä station every 15 minutes. The values of the radar profiles above Hyytiälä are estimated as horizontal
medians over a range of $1\,\mathrm{km}$ from the station. The medians are taken over constant altitudes using linear spatial interpolation
between the rays. The target bin size of the height interpolation is 50 m.

In this investigation, vertical profiles of equivalent reflectivity factor, $Z_\mathrm{e}$, differential reflectivity, $Z_\mathrm{DR}$, and specific differen-
tial phase, $K_\mathrm{dp}$ are considered in the classification. The $K_\mathrm{dp}$ values were computed using the Maesaka et al. (2012) method as
implemented in the Python ARM Radar Toolkit (Py-ART; Helmus and Collis, 2016). The method assumes that propagation
differential phase, $\phi_\mathrm{DP}$, increases monotonically with increasing range from the radar. In this study, we mainly focus on precip-
itation processes typically occuring in stratiform precipitation, where negative $K_\mathrm{dp}$ is not important. The Maesaka et al. (2012)
algorithm should be avoided when studying precipitation events with lightning activity, where negative $K_\mathrm{dp}$ may occur due to
electrification (Caylor and Chandrasekar, 1996). Negative $K_\mathrm{dp}$ has also been reported during events of conical graupel which
have been linked to observations of generating cells (Oue et al., 2015). The total fraction of profiles analyzed in this study
where conical graupel appear or which represent strong convective cells with a possibility for lightning activity, is expected to
be marginal, as discussed further in Sect. 4.

## 2.2 Radar data preprocessing

Prior to training or using the polarimetric radar vertical profile data for the classification, noise and clutter filtering is applied to the binned profiles, which is followed by normalization and smoothing. Additionally, there are different preprocessing procedures for rain and snow events that allow taking ambient temperature into account in the classification. This section describes the mentioned preprocessing steps in more detail.

### 2.2.1 Profile truncation

This paper focuses on identifying, characterizing and investigating the frequencies of different types of vertical structures of dual polarization radar variables specifically from the perspective of detecting, documenting and studying ice processes. Therefore, before the classification, vertical profiles of radar variables are truncated at the top of melting layer (ML), if one is present. Cases where melting layer signatures were not identified and surface air temperature was $1\,°C$ or lower, are placed in the snowfall category and investigated separately.

Following Wolfensberger et al. (2015), who have used gradient detection on a combination of normalized $Z_H$ and $\rho_{hv}$ for ML detection, we combine $\rho_{hv}$ and standardized $Z_e$ and $Z_{DR}$ into a melting layer indicator:

$$I_{ML} = \hat{Z}_e \hat{Z}_{DR}(1 - \rho_{hv}) \tag{1}$$

The same standardization of $Z_e$ and $Z_{DR}$ is used here as in classification, as described in Sect. 3.1. In this study, instead of gradient detection, we use peak detection on smoothed $I_{ML}$ to find the ML. Peaks are defined as any sample whose direct neighbors have a smaller amplitude, and are found in three steps:

1. Peak detection is performed with thresholds for absolute peak amplitude and prominence ($H_{I_{ML}}$; as described below), with chosen values of 2 and 0.3, respectively. The SciPy (Version 1.3; Jones et al., 2019) implementation of the peak detection algorithm[1] is used here.

2. Median ML height, $\tilde{h}_{ML}$, is computed as the weighted median of the peak altitudes, $h_i$, using the product of peak absolute amplitude and $H_{I_{ML}}$ as weights. Peaks above a chosen height threshold of $h_{thresh} = 4200\,m$ are ignored in this step.

3. Step 1 is run again, this time only considering data within $\tilde{h}_{ML} \pm \Delta h_{ML}$ with a chosen $\Delta h_{ML}$ value of $1500\,m$. If multiple peaks exceed the threshold values within a profile, the one with the highest amplitude is used.

The ML top height $h_{ML,top}$ is estimated as the altitude corresponding to the $0.3H_{I_{ML}}$ upper contour of the peak. Peak prominence, $H$, is a measure of how much a peak stands out from the surrounding baseline value and is defined as the difference between the peak value and its baseline. The baseline is the lowest contour line of the peak encircling it but containing no higher peak (Jones et al., 2019).

It should be noted that in steps 2. and 3., the analysis height is limited to reflect the climatology of temperature profiles on the measurement site. In step 2., we assume ML to be always below $h_{thresh}$, and in step 3., we expect melting layer height not to

---

[1]Function scipy.signal.find_peaks

change more than $\Delta h_{\mathrm{ML}}$ during an event. Such use of domain knowledge allows more robust ML detection in situations where $I_{\mathrm{ML}}$ has high values elsewhere. This may occur e.g. in dendritic growth layer (DGL), where the crystals can be pristine enough to cause a significant increase in $Z_{\mathrm{DR}}$ and a decrease in $\rho$.

Sensitivity of the retrieved $h_{\mathrm{ML,top}}$ is tested for small changes in peak detection parameters discarding inconsistent values. A moving window median threshold filter is applied on time series of $h_{\mathrm{ML,top}}$ in order to discard rapid high amplitude fluctuations

caused by e.g. noise in $Z_{\mathrm{DR}}$. A rolling triangle mean with a window size of 5 profiles, correspoding to one hour, is used for smoothing. Finally, linear interpolation and constant extrapolation is applied on $h_{\mathrm{ML,top}}$ on per precipitation event basis to make the estimate continuous. This robust, albeit fairly complex procedure produces a smooth estimate for melting layer top height. The results from the ML detection were analyzed manually and the events with errors were discarded. In 90 % of events in the original data set, ML was detected without errors.

The analysis of rain profiles is limited to a layer from $\Delta h_{\mathrm{margin}} = 300\,\mathrm{m}$ to $10\,\mathrm{km}$ above $h_{\mathrm{ML,top}}$. The purpose of the margin $\Delta h_{\mathrm{margin}}$ is to prevent properties of the melting layer from leaking to the clustering features. The truncation described in this section has no effect on the height bin size.

### 2.2.2 Absence of melting layer

Cutting the rain profiles at the top of melting layer effectively provides information about the ambient temperature at the profile
base. As temperature is a key factor driving the ice processes, such information should be included in the classification process also when there is no ML present. In order to introduce corresponding information on ambient temperature at the profile base, we use surface temperature as an extra classification parameter for events with snowfall on the surface. While it would be possible to use whole temperature profiles from soundings or numerical models as classification parameters, we feel that this may not be feasible for many potential key use cases of the classification method. With the wide availability of surface
temperature observations in high temporal resolution and in real time, presumably this choice makes the classification method more accessible especially for operational applications.

The analysis of snow profiles is limited to a layer between 0.2 km and 10 km above ground level.

## 3 Classification method

The unsupervised classification method used in this study is based on clustering of dual-polarization radar observations, namely
vertical profiles of $K_{\mathrm{dp}}$, $Z_{\mathrm{DR}}$ and $Z_{\mathrm{e}}$. Feature extraction is performed by applying principal component analysis (PCA) on standardized profiles. Clustering is applied on the principal components of the profiles using the $k$-means method (Lloyd, 1982). A flowchart of the whole process is shown in Fig. 1.

While the core method is identical for processing of all radar profiles, information on temperature is included in slightly different way based on if it is raining or snowing on the surface. These differences are explained in sections 2.2.1 and 2.2.2
and highlighted in Fig. 1: For rain events, the profiles are cut at top of melting layer, and for events without a ML, surface

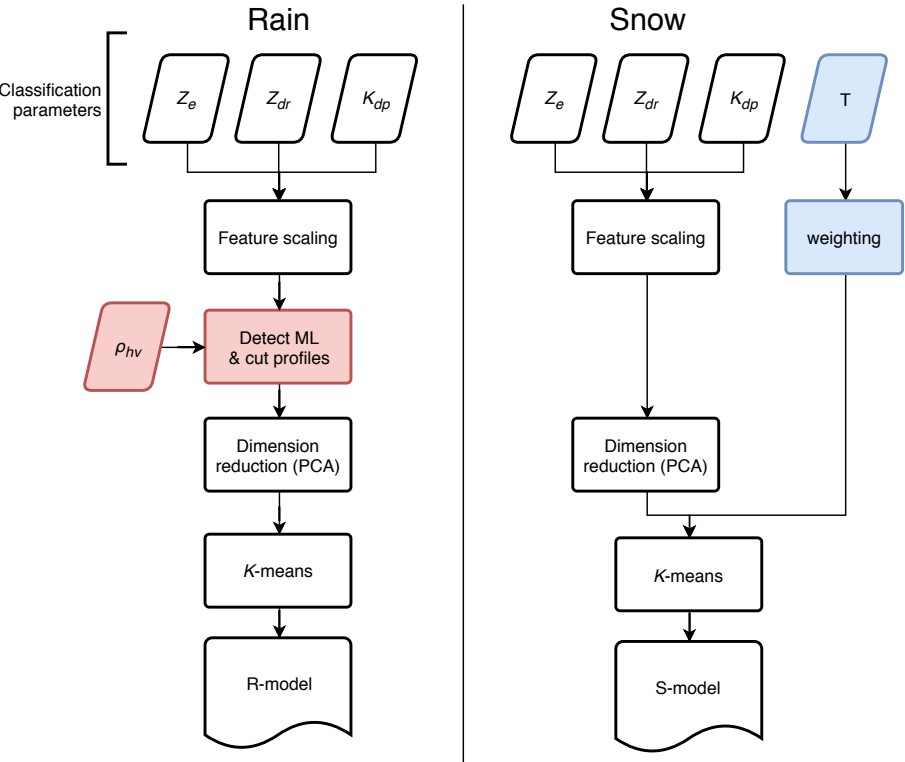

**Figure 1.** Vertical profile clustering method for creating classification models for rain and snow events.

**Table 1.** Standardization of radar variables, $[a, b] \rightarrow [0, 1]$.

| | Rainfall | | Snowfall | |
|---|---|---|---|---|
| | $a$ | $b$ | $a$ | $b$ |
| $Z_e$, dBZ | -10 | 38 | -10 | 34 |
| $Z_{DR}$, dB | 0 | 3.1 | 0 | 3.3 |
| $K_{dp}$, $^\circ\mathrm{km}^{-1}$ | 0 | 0.25 | 0 | 0.11 |

temperature is included as an extra classification variable. Using this approach, information on profile base ambient temperature is included in the classification process, and the analysis is limited to ice processes.

### 3.1 Feature extraction

The vertical resolution of the interpolated data is 50 m with bins from 200 m to 10 km altitude for snow events and from 300 m

to 10 km above melting layer top for rain events. Thus, with the three radar variables, each profile is described by a vector of

588 and 582 dimensions for snow and rain events, respectively. In this study, we apply PCA on standardized profiles of the polarimetric radar variables to extract features for the clustering phase.

A standardization of the preprocessed polarimetric radar data is performed to allow adequate weights for each variable in clustering. This was done separately for the snow and rain data sets in order to account for seasonal differences in the average values. We used similar standardization as Wolfensberger et al. (2015), normalizing typical ranges of values $[a, b] \rightarrow [0, 1]$, with the additional condition that the standardized variables should have approximately equal variances. The values $a, b$ used in this study are listed in Table 1. The values of the standardized variables are not capped, but values greater than 1 are allowed when the unscaled values exceed $b$. Without the standardization, the dominance of each variable in classification would be determined by their variance.

The number of components explaining a significant portion of the total variance for the two training data sets was determined considering the Scree test (Cattell, 1966), the Kaiser method and the component and cumulative explained variance criteria. However, these criteria alone would allow such a low number of components, that the inverse transformation from principal component space to the original would result in unrealistic profiles. Thus, the number of components was increased such that, visually, the inverse transformed profiles presented the significant features in the original profiles, up until to the point where adding more components seemed start explaining trivial features such as noise. For both rain and snow profile classification, the first 30 components are used as features. The high number of significant components suggests that reducing the dimensionality of radar observations is not trivial. An advantage of using PCA over simply sampling the profiles is that the former interconnects data from different heights and radar variables such that the components effectively represent features in the profile shapes, while sampling would rather be driven by absolute values at the individual sampling heights.

With snow profile classification, a proxy of the surface temperature, $P(T_s) = aT_s$, where $a$ is a scaling parameter, is used as an additional feature. Thus, essentially, $\sigma_{T_s}$ within a cluster is decreased with increasing $a$. In this study, the value of $a$ was determined in an iterative process during the clustering phase, described in Sect. 3.2, such that, over the clusters, $\mathrm{median}(2\sigma_{T_s}) \approx 3\,°\mathrm{C}$. Thus, assuming $T_s$ is normally distributed within a given cluster, approximately 95 % of the values of $T_s$ would be typically within a range of $3\,°\mathrm{C}$ from the cluster mean. A value of $a = 0.75$ was used in this study.

## 3.2 Clustering

In the present study, the widely used $k$-means method was chosen for clustering. The algorithm is known for its speed and easy implementation and interpretation. Limitations of the method include the assumption of isotropic data space, sensitivity to outliers (Raykov et al., 2016), and the possibility to converge into a local minimum which may result in counterintuitive results. In our analysis, the anisotrophy of the data space is partly mitigated by the PCA transformation. After the transformation, there is still anisotrophy, but the transitions in density of the data points in PCA space are smooth (not shown), such that $k$-means seems to produce clusters of meaningful sizes and shapes. The problem of local minima is addressed using the $k$-means++ method (Arthur and Vassilvitskii, 2007) to distribute the initial cluster seeds in a way that optimizes their spread. The $k$-means++ is repeated 40 times and the best result in terms of sum of squared distances of samples to their closest cluster center is used for seeding.

## 3.3  Selecting the number of classes

An important consideration in using $k$-means clustering is the choice of number of clusters, $k$. A good model should explain the data well while being simple. Several methods exist for estimating the optimal number of classes. Nevertheless, often domain and problem specific criteria have to be applied for the best results.

The optimal number of clusters depends on variability in the data and correlations between different variables. The more variability and degrees of freedom, the more clusters are generally needed to describe different features in a dataset. Since one important use case for the method is ice process detection, particular attention is paid in separation of fingerprints of different processes between classes. An optimal set of classes would maximize this separation without introducing too many classes to make their interpretation complicated.

As the problem of the number of classes is complex, it is difficult to find an unambiguous quantitative measure for evaluating the correct number of classes. Attempts to create a scoring function for optimizing the separation of ice processes alone did not yield satisfactory results, but were rather used to support the manual selection process.

Silhouette analysis (Rousseeuw, 1987), which is a method for measuring how far each sample is from other clusters (separation) compared to its own cluster (cohesion), was also considered for selecting $k$. The metric, silhouette coefficient $s$, takes values between -1 and 1. The higher the value, the better the profile represents the cluster it is assigned to. A profile with $s = 0$ would be a borderline case between clusters, and negative values indicate that the profiles might have been assigned to wrong clusters. Silhouette score $\bar{s} = \frac{1}{k}\sum_{i=1}^{k} s_i$ can generally be used for choosing $k$. Unfortunately, when applied to the radar profile clustering results, $\bar{s}$ decreases almost monotonically with increasing $k$ in the ranges of $k$ analysed, and thus did not prove very useful for this purpose. Rather, in this study, we calculate $s$ for each profile classification result individually as a measure of how well the profile represents the class it is assigned to.

The process of selecting the number of rain and snow profile clusters, $k_R$ and $k_S$, respectively, was as follows: First, the $k$-means clustering was repeated 12 times for each $k$ in $[5, 21]$ with 40 $k$-means++ initializations. This is where the above-described silhouette analysis was performed for each set of clusters and the stability of the initialization process was analyzed for each $k$. Between the 12 repetitions, the clustering converges to identical results for each $k_R < 12$ and $k_S < 10$. With higher values of $k$, there are multiple solutions to the clustering problem with only minor differences between them. Moreover, the properties of the cluster centroids are not highly sensitive to $k$. Clustering performed with $k = k_0$ and $k = k_0 + 1$ would typically result in sets of clusters sharing $k_0 - 1$ to $k_0$ very similar centroids.

This stability of the clustering results makes it convenient to select $k$ manually. In the second stage, we analysed each separate clustering solution for differences between the clusters from the point of view of snow processes and surface precipitation. Specifically, an important criterion was to separate the typical $K_{dp}$ signatures of dendritic growth (e.g. Kennedy and Rutledge, 2011) and the H-M process (Field et al., 2016) into different classes. On the other hand, the use of an unsupervised classification method should also allow us to discover previously undocumented features in the radar profiles if they are present in the data in significant numbers.

The goal in this step is to find as many significant unique fingerprints with as low $k$ as possible by manual evaluation. Significant differences between clusters in this context include variations in profile shapes and altitudes of characteristics such as peaks, clear differences in echo top heights, or differences of cluster centroid $T_s$ of more than $3\,°C$. The most common trivial difference between a pair of clusters was a difference in the intensity of polarimetric radar variables while the shapes of the cluster centroid profiles were almost identical. Altitude differences between fingerprints of overhanging precipitation were also considered trivial.

During this process, allowing some profile classes with only trivial characteristics was inevitable in order to include others with significant unique fingerprints. For this reason, some classes likely reflect natural variatiability of the same microphysical process rather than unique processes, and need to be combined. However, the optimal way of combining the classes may depend on the application. Thus, we present the classes uncombined in this paper.

In snow profile clustering, $T_s$ as an extra classification parameter adds a significant additional degree of freedom. Thus, a larger number of snow profile classes are needed to meet the criteria described above. In clustering, there is a distinguishable separation between clusters representing $T_s$ close to $0\,°C$ and around $-10\,°C$. The vast majority of profiles belong to the warmer group.

Taking all the mentioned considerations into account, we chose to use 10 and 16 classes for rain and snow profiles, respectively. In 12 of the snow profile class centroids, $T_s > -5\,°C$. In this paper, the rain and snow profile classification models are termed R-model and S-model, respectively.

In this section we have described our approach for optimizing the number of classes with the main criteria of separating the main profile characteristics and the fingerprints of ice processes into individual classes. It should be noted, however, that there is a large spectrum of research problems and operational applications where an unsupervised profile classification method such as the one described in this paper could be potentially useful. The optimal number of classes may depend on the application.

## 4 Results

Class centroids of rain and snow profile classes are shown in Figs. 2 and 3, respectively. The centroid profiles of dual polarization radar variables are inverse transformed from corresponding centroids in PCA space. Classes are numbered in the ascending order by the value of the first principal component in the class centroids. By definition, the first component has the largest variance and has therefore the biggest influence on the clustering and classification results. The value of this component is strongly correlated with intensities of $K_{dp}$ and $Z_e$.

A number of class centroids in both classification models display distinct features in dual polarization radar variables often linked to snow processes, such as peaks and gradients in $K_{dp}$ and $Z_{DR}$. Such features and their connection to other characteristics in the vertical structure of the profiles, and finally to the precipitation processes are discussed in this section.

As a general pattern in Figs. 2 and 3 we see that the highest values of $Z_{DR}$ are associated with low echo tops while the highest $K_{dp}$ values occur in deeper clouds. This is in line with the previously reported findings (Kennedy and Rutledge, 2011; Bechini et al., 2013; Moisseev et al., 2015; Schrom et al., 2015; Griffin et al., 2018) that echo tops in DGL are associated with

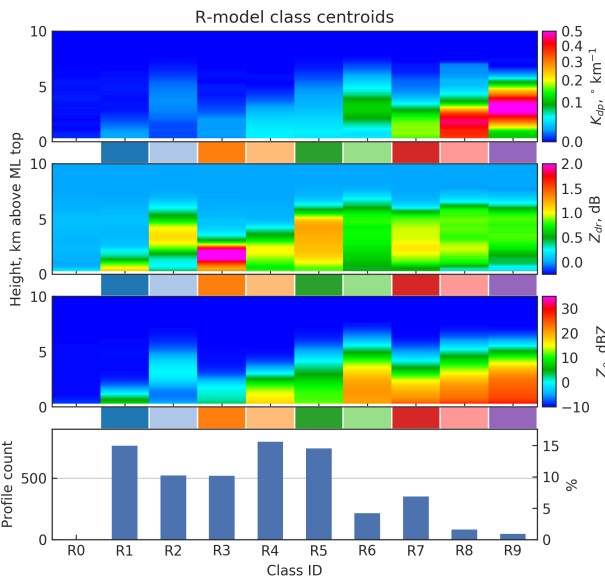

**Figure 2.** Class centroid profiles of the R-model. Profile counts per class are shown at the bottom omitting the count for low-reflectivity class R0. Between the panes, each class has been assigned a color code.

high $Z_{DR}$ and low $K_{dp}$ in the layer, whereas high $K_{dp}$ in the DGL with low $Z_{DR}$ is associated with echo tops in $T < -37\,°C$ where homogeneous freezing occurs. Using the NCEP GDAS model output, we analyzed the echo top temperatures, $T_{top}$, of each vertical profile radar observation. The results, grouped by profile class, are visualized in Fig. 4. It should be noted, that in the summer, cold echo tops may be caused by strong updrafts in convection, whereas during the winter, echo tops colder than

approximately $-37\,°C$ are a more unambiguous indication of homogenous freezing. Inspecting the class centroids in Figs. 2 and 3, and comparing them to echo top heights in Fig. 4, it is evident that $K_{dp}$ layers, especially elevated ones, are strongly associated with high echo tops.

The clustering results expose a prominent seasonal difference in $K_{dp}$ intensity: consistently lower values are present in snow events. There are 4 rain profile classes in contrast to only 2 snow profile classes with peak cluster centroid $K_{dp}$ exceeding

$0.1\,°km^{-1}$. They represent total fractions of 13 % and 4 % of rain and snow profiles, respectively. Corresponding to this difference, in Figs. 2 and 3, as well as in Figs. 7 and 8 introduced later, $K_{dp}$ is visualized in different ranges in relation to rain and snow profiles. The seasonal differences in $Z_{DR}$ and $Z_e$ intensities are less prominent. High $K_{dp}$ in the summer may be linked to higher water content during the season. Additionally, the seasonal variability of vertical motion could impact the $Z_{DR}$ and $K_{dp}$ enhancements.

Convection in the summer, especially in the presence of hail, is linked to extreme values of radar variables and high echo tops (Voormansik et al., 2017), which may also have a small contribution to the seasonal differences (Mäkelä et al., 2014). However, convective rain storms are of short duration, and thus typically present in just a couple of profiles per a convective cell. Therefore, their impact on the class properties are expected to be limited. Manual analysis revealed that classes R6 and

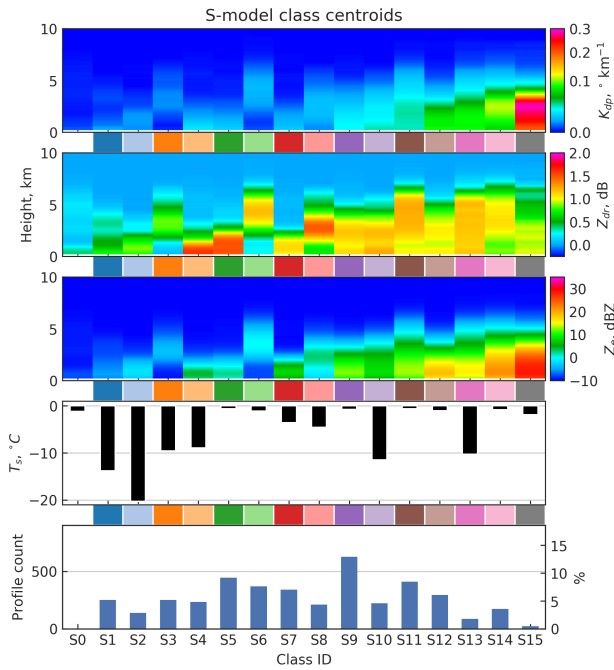

**Figure 3.** Class centroid profiles of the S-model. Top panel shows class centroid surface temperatures. Profile counts per class are shown at the bottom omitting the count for low-reflectivity class S0. Between the panes, each class has been assigned a color code.

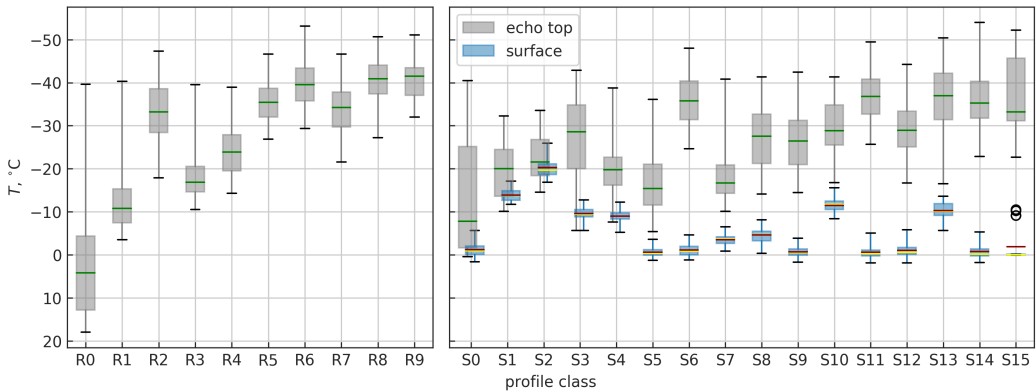

**Figure 4.** Cloud top temperature distributions by class (gray) with green line marking the medians. For S-classes, also surface temperature distribution is shown (blue) with red lines marking the class centroid and yellow lines marking the median. Boxes extend between the 1st and the 3rd quantiles, and whiskers cover 95 % of the data.

R9 have the highest, and R5 the lowest fractions of profiles measured in convective cells. Further details of this analysis are presented in Sect. 4.2.

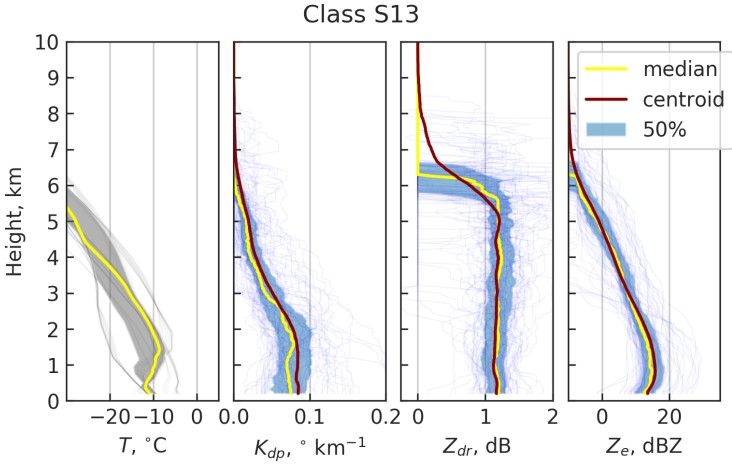

**Figure 5.** Class S13 centroid is visualized on the three rightmost panes. Individual class member profiles are marked with thin lines. The pane on the left shows corresponding NCEP GDAS temperature profiles. The areas between the first and the third quantiles are shaded, radar data in blue and GDAS in gray.

Class frequencies are presented in the bottom panels of the Figures 2 and 3. Classes S0 and R0 represent very low values of $Z_{\mathrm{e}}$ throughout the column, i.e. profiles with very weak or no echoes. Therefore their frequencies depend merely on the subjective selection of observation period boundaries, and are thus omitted in the figures. Boundaries of the precipitation events are partly based on these two 0-classes. Events are considered independent and separate if between them there are profiles classified as

S0 or R0 continuously for at least 12 hours.

With respect to $K_{\mathrm{dp}}$ intensity, classes in the R-model can be divided into four categories: R0 through R3 with negligible $K_{\mathrm{dp}}$, low-$K_{\mathrm{dp}}$ classes R4 and R5 with $\max(K_{\mathrm{dp,c}}) \approx 0.04\,^{\circ}\mathrm{km}^{-1}$, high-$K_{\mathrm{dp}}$ classes R6 and R7 with $\max(K_{\mathrm{dp,c}}) > 0.11\,^{\circ}\mathrm{km}^{-1}$, and classes R8 and R9 representing extreme values $(\max(K_{\mathrm{dp,c}}) \approx 0.5\,^{\circ}\mathrm{km}^{-1})$. The subscript "c" denotes a class centroid value as opposed to values in individual profiles. The peak $K_{\mathrm{dp,c}}$ of both R6 and R9 is at 3 km, corresponding to class mean GDAS

temperatures of -16 °C and -18 °C, respectively. Essentially, these two classes represent clear $K_{\mathrm{dp}}$ features in the DGL.

Classes R7 and R8 feature considerable $K_{\mathrm{dp}}$ in 2–3 km thick layers right above ML, with centroid values slightly below $0.2\,^{\circ}\mathrm{km}^{-1}$ and around $0.4\,^{\circ}\mathrm{km}^{-1}$, respectively. Essentially, both classes represent $K_{\mathrm{dp}}$ signatures in both DGL and temperatures favored by the H-M process. Sinclair et al. (2016) found that the typical $K_{\mathrm{dp}}$ values for the H-M process are capped at $0.2$–$0.3\,^{\circ}\mathrm{km}^{-1}$ for C-band due to onset of aggregation. Based on this, it can be argued that R7 is a more likely indicator of

H-M than R8.

Classes R3 and R4 were found to often coexist in precipitation events. Both are characterized low $K_{\mathrm{dp}}$ and a layer of $Z_{\mathrm{DR}}$ in DGL. In Fig. 4, we see that the echo tops are lower for the R3 profiles, typically in the DGL. Therefore, we would expect growth of pristine crystals in low number concentrations, and consequently with no significant aggregation. This would explain why peak $Z_{\mathrm{DR}}$ values from 3 to $5\,\mathrm{dB}$ are common in relation with R3. Profiles classified as R4, on the other hand, have slightly

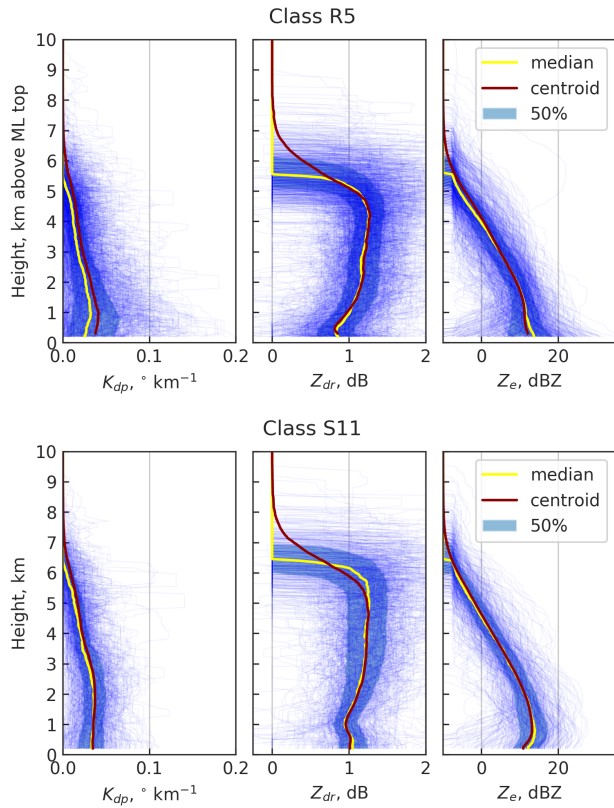

**Figure 6.** Comparing classes R5 (top panels) and S11 (bottom panels) shows evident similarities. Individual class member profiles are marked with thin blue lines and the areas between the first and the third quantiles are shaded with blue.

higher echo tops ($T < -20°$C), which are expected to result in higher number concentrations, leading to aggregation. The R4 profiles are characterized by much lower $Z_{DR}$ values.

In the S-model (Fig. 3), classes S0 through S3 represent profiles with low values of all three radar variables, each with $\max(Z_{e,c}) < 0$ dBZ, $\max(Z_{DR,c}) < 1$ dB and $\max(K_{dp,c}) < 0.01°$km$^{-1}$. These four low reflectivity classes represent different surface temperatures, which is likely a major driver for the separation of these classes in the clustering process. Classes S4 and S5 represent low echo top profiles with high $Z_{DR}$, with class centroid surface temperatures of $-9.0°$C and $-0.6°$C, respectively. Further analysis of NCEP GDAS temperature profiles reveals that, across the board, there is an inversion layer present where radar profiles are classified as S4, typically with temperatures below $-10°$C within the lowest kilometer. This corresponds well with the bump in $Z_{DR,c}$ close to the surface, suggesting possible growth of pristine dendrites within a strong inversion layer. In contrast, there is no inversion in connection with profiles belonging to S5, and the enhancement in $Z_{DR}$ occurs already at 2 to 3 km above the surface, where the median NCEP GDAS temperature for S5 profiles is roughly between $-18$ and $-10°$C. S5 is the second most common class in S-model classification results.

Classes S6 and S8 represent situations where precipitation is detached from the surface. These types of profiles are typically present in association with approaching frontal systems before the onset of surface precipitation. The most frequent class of the S-model is S9 covering 13 % of the profiles. It represents moderate values of polarimetric radar variables and cloud top height. The most extreme values of reflectivity and $K_{dp}$ values in the S-model are represented by classes S14 and S15. For both classes, $K_{dp,c}$ peaks above $3\,km$ suggesting dendritic growth in the member profiles. Values of $Z_{DR,c}$ are significantly lower compared to other high echo top classes with weaker $K_{dp,c}$. Class S15 can be seen as a more extreme variant of S14 with much stronger $K_{dp,c}$ and $Z_{e,c}$. In addition, S15 represents lower values of $Z_{DR}$ near the DGL, having slightly elevated values in the bottom three kilometers instead. These differences are likely due to even higher ice number concentrations in S15 profiles, which lead to more intense aggregation.

Comparing class centroid $T_s$ and class frequencies in Fig. 3 it can be seen that most snowfall occurs at $T_s \approx 0\,°C$. Further analysis of GDAS temperature profiles for the snow events revealed that typically cold surface temperatures ($T_s < -6\,°C$) are heavily contributed by strong inversion layers. The centroid and members of S13 are visualized in Fig. 5, along with the member GDAS temperature profiles. The profile class is characterized by a thick layer of considerable $K_{dp}$ from 2 to 3 km to the surface, and $T_s \approx -10\,°C$. As seen in the left panel of Fig. 5, S13 represents conditions where $T$ typically falls below $-10\,°C$ close to the surface. This finding suggests that a second DGL may occur in a strong inversion layer.

Using the double-moment Morrison microphysics scheme (Morrison et al., 2005), Sinclair et al. (2016) showed that $K_{dp}$ at the $-8$ to $-3\,°C$ temperature range can be used for identifying the H-M process. Such fingerprints are present especially in profiles classified as R7 or S12. However, manual analysis of the profile data revealed that both of these classes represent a mixture of fingerprints indicating H-M, dendritic growth or the co-presence of both processes. In several events, there were continuous time frames of profiles classified as either R7 or S12 during which the altitude of the $K_{dp}$ signal was changing from profile to profile between DGL and $0\,°C$ level, and was occasionally bimodal. One example of such time frame is shown in Fig. 7 and discussed further in Sect. 4.1.1. Some bimodality is present also in the centroid $K_{dp,c}$ of both classes, suggesting that the elevated $K_{dp,c}$ values in the H-M region cannot be explained solely by sedimenting planar crystals generated aloft, but are contributed by the H-M process.

While neither in rain nor snow profile classification, there are classes with clear-cut $K_{dp,c}$ peaks at altitudes corresponding to temperatures preferred by the H-M process, there are, in contrast, several classes with strong elevated $K_{dp,c}$ layers. The proposal of Sinclair et al. (2016) that $K_{dp}$ fingerprints of the H-M process are not very pronounced may explain the tendency of the classification method not to produce more pure H-M classes. Nevertheless, R7 and S12 can be used as indicators for conditions where H-M may occur.

Despite the differences in the classification methods for rain and snow profiles, there are prominent similarities between the two models and profile classes therein. Archetypal classes such as high echotops in the presence of elevated $K_{dp}$ layers (R6, R9, S14, S15) or high $Z_{DR}$ in shallow precipitation (R3, S4, S5) exist in both classification models. Frequent classes R5 and S11, visualized side by side in Fig. 6 can be considered direct counterparts of each other; The vertical structure of polarimetric radar variables above ML in R5 match strikingly well with S11. The two classes are characterized by weak $K_{dp}$ and typical

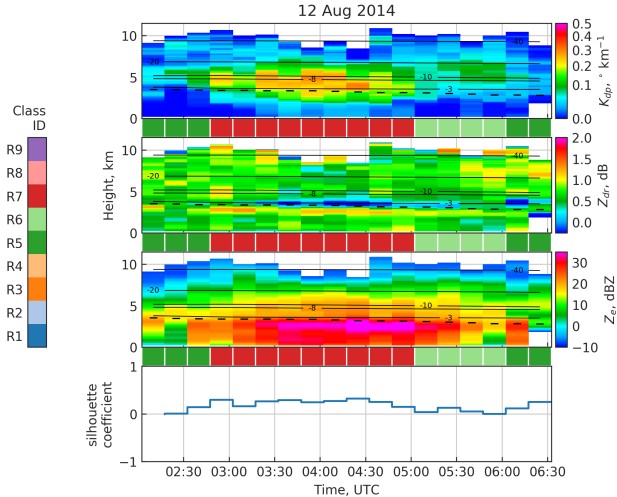

**Figure 7.** Classification analysis of a rain case with silhouette scores. The automatically detected melting layer is marked with a dashed line, the solid lines show the NCEP GDAS temperature contours and the colors between the panes denote classification results.

values of $Z_{DR}$ slightly above 1 dB aloft, decreasing towards the altitude corresponding to $0\,°C$. Presumably, this indicates the presence of aggregation.

## 4.1 Case studies

In Figures 2 and 3, each class is assigned a color code (between the panels). This color coding is used in Figures 7 and 8 to
mark classification results in a rain and a snow case, respectively. Note, that the same set of colors are used for denoting rain and snow profile classification, but they should not be confused with each other.

### 4.1.1 August 12, 2014

In Fig. 7, rain profile classification has been applied on a precipitation event from August 12, 2014. During this event, echo tops repeatedly exceed $10\,\mathrm{km}$. Only the parts of the profiles above melting layer top are analyzed here, since everything below
that level is invisible to the classifier. The first two and the last two profiles shown in the figure are characterized by low $Z_e$ and low $K_{dp}$, while $Z_{DR}$ has values around $1\,\mathrm{dB}$. These profiles are classified as R5 (dark green). Between 2:30 and 3:00 UTC, a significant increase in $K_{dp}$ occurs followed with an increase in reflectivity and decrease in $Z_{DR}$. The temperature (altitude) of the downward increase in $K_{dp}$ varies from $-20\,°C$ level to closer to ML. In this phase, there is also a small increase in $Z_{DR}$ in the DGL whenever the increase of $K_{dp}$ also occurs in the DGL. This phase in the event is sustained until around 5:00 UTC
and is classified as R7 (dark red). It is followed by approximately an hour of a weaker elevated $K_{dp}$ layer at around 4 to $6\,\mathrm{km}$ altitude with profiles classified as class R6 (light green). The silhouette coefficient is positive throughout the event indicating

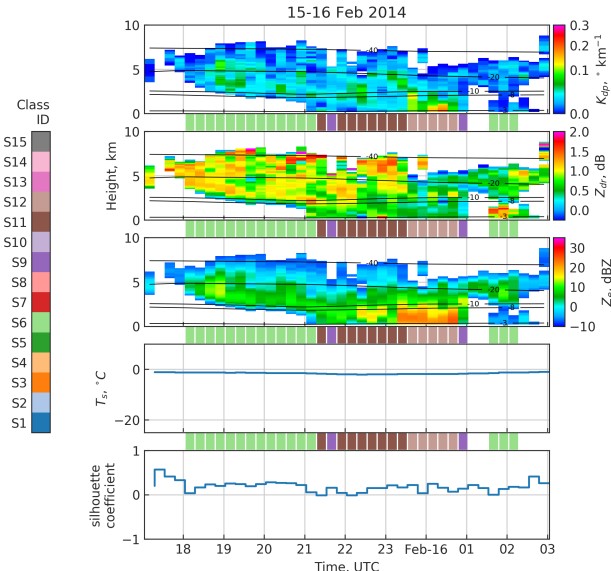

**Figure 8.** Classification analysis of a snow case with silhouette scores. Solid lines show the NCEP GDAS temperature contours and the colors between the panes denote classification results.

good confidence of the classification results. The silhouette of the profiles classified as R6 is not very high, though, which is likely due to lower values of $Z_e$ compared to the class centroid.

Similar analysis of more rain events in the data set reveals that, similarly to the August 12 event, R7 typically coincides with an increase of $K_{dp}$ in the DGL, H-M layer or both, often with varying altitude. Without in situ observations or analysis of Doppler spectra, it is not trivial to tell whether this variability is due to co-presence of dendritic growth and H-M, or simply fall streaks. Class R6, on the other hand, is more specific to a $K_{dp}$ fingerprint in the DGL. The more infrequent profiles with clear $K_{dp}$ bands above the DGL are typically also classified as R6 or R9.

### 4.1.2 February 15–16, 2014

Classification results for February 15–16, 2014 are shown in Fig. 8. The event has a clear structure of an approaching frontal system. Between 17 and 18 UTC $Z_e$ is very low, corresponding to class S0, which is marked with white color between the panels. Between 18 and 21 UTC, the event starts with overhanging precipitation, classified as S6 (light green). This is followed by light precipitation with echo tops roughly 7 to 8 km, and relatively high $Z_{DR}$ near the echo top, decreasing downwards. This corresponds well with class S11 (dark brown). After 23:30 UTC, The echo top height is decreased to roughly 6 km, $Z_{DR}$ is decreased and $K_{dp}$ signals appear close to ground level. The increase in $K_{dp}$ occurs within the -8 to $-3\,°C$ temperature range suggesting the presence of the H-M process. Indeed, Kneifel et al. (2015) report needles, needles aggregates and rimed particles on the surface at the measurement site during this period, and favorable conditions for rime splintering. Further, using Weather Research and Forecast (WRF) model, Sinclair et al. (2016) showed that secondary ice processes are needed to explain

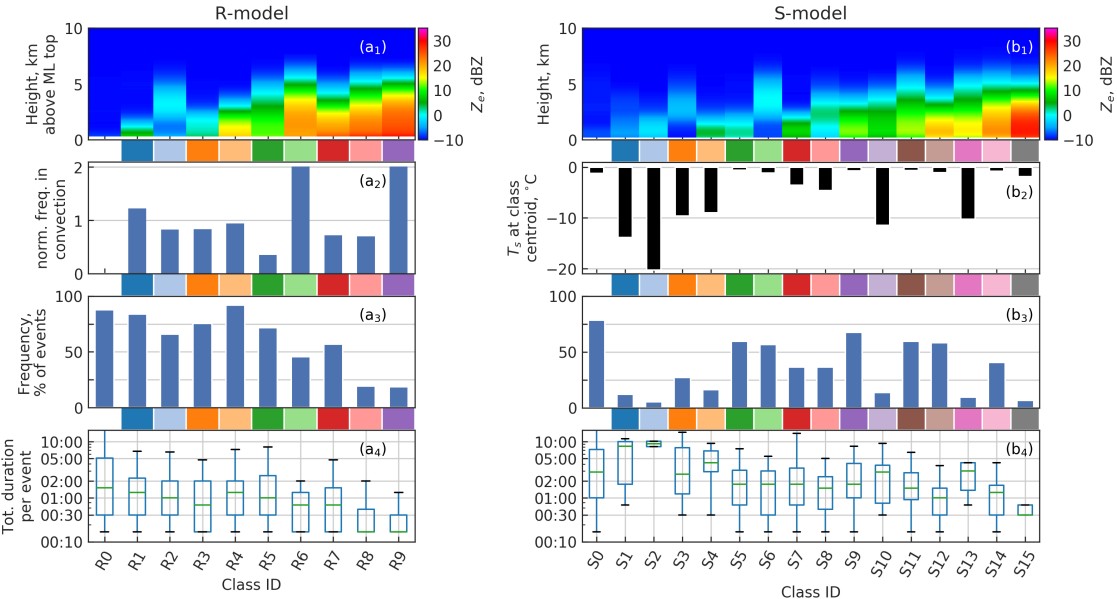

**Figure 9.** Statistics on frequency of each profile class. Classes are identified by class centroid $Z_e$ at the top panels, class centroid $T_s$ for snow profile classes ($b_2$), color codes between the panels and class IDs at the bottom. Panel ($a_2$) has the ratios of the number of profiles in convective cases per class to the expected value in uniform distribution. In the bottom two panels, the class frequencies are given as percentage of events ($a_3$, $b_3$) and total durations ($a_4$, $b_4$) within events.

the observed number concentrations during this time period. The corresponding profiles are classified as S12 (light brown).

Within this case study, two profiles, marked with dark purple color, are classified as S9, likely due to the momentary absence of any strong $K_{dp}$ or $Z_{DR}$ signals.

## 4.2 Statistics

Frequency statistics of the profile classes are presented in Fig. 9. We analyzed a subset of rain events as either convective or stratiform using a number of sources of publicly available satellite and numerical model data. Out of 70 events analyzed, 15

were convective and 55 stratiform. Panel ($a_2$) in Fig. 9 shows the ratios of the number of profiles in convective cases per class to the expected value in uniform distribution. On average, twice as many profiles are classified as R6 and R9 in convective situations compared to their average frequencies. Both classes are characterized by high echo tops and elevated $K_{dp}$ bands. On the other hand, classes R7 and R8, also representing high $K_{dp}$ values, but closer to the melting layer than R6 and R9, appear in lower than average frequency in convective situations. Class R5 is most pronouncedly characteristic for stratiform events, with

frequency in convective events roughly one third of the average value.

Panels ($a_3$) and ($b_3$) of Fig. 9 show the fractions of independent precipitation events in which each class occurs. With rain events, this frequency correlates inversely with $K_{dp,c}$. Rain profiles classified as R8 and R9, which represent the strongest $K_{dp}$

signatures, occur in 20 % and 19 % of the events, respectively, with at least one of the two occurring in 25 % of the events. Classes R6 and R7, representing more modest $K_{dp}$ features, occur in 45 % and 57 % of cases, respectively, and the rest of the

classes between 67 % and 92 % of the cases.

With snow events, the likelihood of a given class occurring within an event correlates not only with peak $K_{dp,c}$ but also with surface temperature. Any class representing low $K_{dp}$ values and surface temperature close to 0 °C occurs in more than half of the snow events.

The per precipitation event class persistence is visualized in the bottom panels of Fig. 9. Profile classes representing the

highest values of $Z_e$ at the surface, namely R6–R9, S12, S14 and S15, are short-lived, whereas snow profile classes character-ized by cold surface temperatures are the most persistent. Profiles classified as R0 or S0 omitted, the median durations of rain and snow events in the data set are 5.5 h and 11.5 h, respectively. This difference explains why S-classes are on average more persistent than R-classes.

## 5    Conclusions

A novel method of dual polarization radar profile classification for investigating vertical structure of snow processes in the profiles was presented in this paper. The method is based on clustering of PCA components of vertical profiles of $K_{dp}$, $Z_{DR}$ and $Z_e$, and surface temperature. It was applied on vertical profile data extracted from C-band RHI scans over Hyytiälä mea-surement station in Southern Finland. We applied separate versions of the method based on if surface precipitation type was rain (R-model) or snow (S-model). In the R-model, profiles are truncated at the melting layer top, and in the S-model, surface

temperature is used as an additional classification feature. The content of the vertical profile classes was manually interpreted.

In the present investigation, some class centroids resembled textbook examples of previously documented snow process fingerprints, while others may represent a mixture of different conditions. If temperature profiles from either soundings or numerical models are available, the interpretation can be done in the absence of surface crystal type reports. Notably, this is prequisite in cases of rainfall when direct observations of crystal types cannot be performed at the surface.

The year-round variability in the vertical structure of $K_{dp}$, $Z_{DR}$ and $Z_e$ can be described using a total of 26 profile classes; 10 and 16 in the presence and absence of ML, respectively. One of the main goals of this study was to associate profile classes with snow processes for their automated identification. It should be noted, though, that the profile classification is not based on expressly selected characteristics of radar fingerprints of the processes, but rather the general, complete structure of the profiles. Nevertheless, some profile classes seem to be strong indicators of specific processes or their combinations within the

vertical profiles. From both classification models we can identify a total of 7 archetypes with the following characteristics:

1. Strong $K_{dp}$ band in DGL, while $Z_{DR}$ band is not pronounced. Deep precipitation system with homogeneous freezing at the cloud top. Associated with intensified dendritic growth leading to aggregation and high precipitation rate. (Classes R6, R9, S14, S15)

2. $K_{dp}$ signature between DGL and 0 °C level possibly due to simultaneous occurrence of dendritic growth and secondary

ice production. Homogeneous freezing at the cloud top. (R7, R8, S12)

3. High echo top, negligible $K_{dp}$, and $Z_{DR} > 1$ dB, which decreases closer to the melting level due to aggregation. Typically, $Z_e < 20$ dBZ. (R5, S11)

4. Cloud top between -30 and -20 °C level, with only a weak $Z_{DR}$ band at -15 °C level. Moderate $Z_e$ of roughly 20–30 dBZ, and weak $K_{dp}$. (R4, S9)

5. Strong $Z_{DR}$ of typically more than 1.5 dB at the cloud top at around -15 °C level associated with growth of pristine planar crystals in low number concentrations. No $K_{dp}$ is present, and low values of $Z_e$ indicate the absence of aggregation. (R3, S5)

6. Echo detached from the surface due to snow particles either not having reached the surface yet or sublimating due to a dry layer. (R2, S3, S6, S8)

7. Weak or no $Z_e$. (R0, S0–S2)

In addition to these archetypes found in both summer and winter storms, there are S-classes representing situations where strong inversions interfere with snow processes. Notably, we found indications of dendritic growth in strong inversion layers, manifested as class S13. As the colder arctic air mass seldom occurs in Southern Finland, $T_s < -10°C$ can usually be attributed to a strong lower level inversion. Such inversions may have an important effect on the frequency of occurrence of some ice processes. Further, this implies that temperature information near the surface is necessary in order to determine whether a low altitude $K_{dp}$ signature in the winter is an implication of the H-M process or dendritic growth.

Our approach to the classification problem is pronouncedly data-driven. This way, if the training material represents the climatology of ice processes and their radar signatures, as was the aim in this study, the resulting classes will reflect the statistical properties of this climatology. Hand picking the training material, on the other hand, would introduce human bias to the class boundaries.

However, there are possible drawbacks in the data-driven approach. The typical radar fingerprints of the H-M process were found to be much more scarce than those of dendritic growth, and often less pronounced. This negatively affects how the typical fingerprints of H-M process are represented in the classes. This could be enhanced by introducing a larger fraction of H-M profiles in the training data.

Another disadvantage in the data-driven approach is that covering a meaningful collection of unique fingerprints requires a large number of clusters, some of which do not represent unique microphysical processes. This problem may be mitigated to some extent by further optimizing the scaling of the radar variables such that the clustering would be less driven by differences in the intensities of the signatures in contrast to their shapes. Another way to address this issue is to simply combine classes that seem to represent the same processes, in like manner of the archetypes presented above. Reducing the number of classes by simply choosing a smaller $k$ in the $k$-means clustering would reduce the amount of manual work involved in defining the class boundaries at the cost of decreased detail and accuracy in separating the processes. With a smaller $k$, the clustering would be driven by more general features of the profiles such as the overall shape and intensity of the polarimetric radar variables, whereas especially the typical characteristics of the H-M process fingerprints involve a higher level of detail.

The classification method presented in this study should be considered a starting point in studying vertical profiles of radar variables using unsupervised classification. As such, there is a vast range of potentially useful opportunities for further development of the method. The method is built on reasoned use of well known, proven algorithms such as PCA and $k$-means. We showed that this combination of machine learning algorithms allows both identification of known fingerprints and a more explorative approach in studying the characteristics of a regional climatology of precipitation processes. Limitations of the $k$-means method include the spherical shape and similar area occupied by the clusters, which involves a risk of suboptimal separation of the microphysical processes to different classes. In this study, we addressed these limitations by allowing a rather large number of initial classes and combining similar ones by identifying the archetypes based on known fingerprints of the processes. Another possible approach would be to explore the numerous alternative clustering methods for a more optimal separation of the precipitation processes. A comprehensive comparison of such methods, however, is outside the scope of this study.

In the present classification method, ambient temperature is known only at the profile base. Compared to the use of full temperature profiles, this simplifies the method, and perhaps even more importantly, the requirements for the input data. However, future studies should investigate if the use of full temperature profiles allows more accurate separation of precipitation processes into different classes.

The unsupervised nature of the classification method is expected to allow extending its application to the detection of ice processes not covered in this study. Recently, Li et al. (2018) showed that certain combinations of $Z_\mathrm{e}$, $Z_\mathrm{DR}$, and $K_\mathrm{dp}$ signatures can potentially be used for detecting heavy riming. Furthermore, the process is frequently observer if Finland, highlighting the potential of using an unsupervised method for its identification.

It should be noted that wind shear effects induce differential advection of hydrometeors at different altitudes affecting the gradients in the vertical profiles of radar variables (Lauri et al., 2012). Therefore caution should be used in interpreting microphysical processes corresponding to class centroid profiles. The wind shear effects are difficult to correct for using vertical profile or RHI radar observations due to the limitations in horizontal sampling. Such adjustments become more viable if classification is performed on profiles extracted from volume scans, which will be investigated in future work.

The ability to describe a climatology of vertical stucture of dual polarization radar variables and, further, precipitation processes using a finite number of classes has evident potential in improving quantitative precipitation estimation. We anticipate that automated detection of ice processes may allow the development of adaptive relation for snowfall rate $S = S(Z_\mathrm{e})$, in which the parameters could be chosen based on the profile classification result. Adaptive $S(Z_\mathrm{e})$ relations, in turn, have potential in improving vertical profile of reflectivity correction methods. Future work will be devoted to investigating the use of unsupervised profile classification in such applications.

*Data availability.* The FMI radar and surface temperature data are available from the Finnish Meteorological Institute open data portal: https://en.ilmatieteenlaitos.fi/open-data-sets-available.

*Competing interests.* The authors declare that they have no conflict of interest.

*Acknowledgements.* We would like to thank the personnel of Hyytiälä station and Matti Leskinen for their support in field observation. The research of JT and DM was supported by the Academy of Finland Finnish Center of Excellence program (grant 307331). JT acknowledges the Doctoral Programme in Atmospheric Sciences (ATM-DP, University of Helsinki) for financial support.

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
