# Peer review of "Unsupervised classification of vertical profiles of dual polarization radar variables"

_Atmospheric Measurement Techniques, 2019_

## Referee Comment (RC1) · Anonymous Referee #1 · 11 Sep 2019

This paper presents an attempt at objectively classifying vertical profiles of polarimetric variables in the solid precipitation medium. It then tries to establish links between the resultant classification and known fingerprints of microphysical processes in the upper layer of the atmosphere. The methodology proposed is quite novel and it may open the door to move from the interpretation of the data by an expert to the automatic extraction of relevant features from the data. Moreover, the same methodology may be applied to the study of other phenomena. The paper is clear and well written. I recommend its publication provided some issues I expose below are addressed.

General comments:

1. Generally speaking the paper is well written but it is clear that the authors do not have English as their mother language. In particular, oftentimes articles are missing in the sentence. This does not hinder the understanding of the text so I will not go through the list of what I have spotted but a conscious correction should be performed prior to publication

2. My main concern is with the method used for the clustering. To my understanding, the k-means algorithm expect all clusters to be of similar size. This is an unreasonable assumption in the case of weather phenomena since there are processes that are fairly common whereas others happen rarely. The authors, rightly, do not make any a-priori attempt to balance the data but I suspect that leads to classes that are a mix of various phenomena and hence difficult to link to specific microphysical processes. Other clustering methods such as Expectation Maximization (EM) clustering may be more adequate for the sort of data at hand.

3. Given that microphysical processes in the solid phase of precipitation are highly dependent on the ambient temperature and the authors have available an estimate of the temperature profile via the NCEP GDAS I would be very interested in having a look at the results of the clustering when including the full temperature profile instead of just the surface temperature.

4. I am a bit surprised by the choice of algorithm to compute KDP. The Maesaka algorithm targets primarily the liquid layer of precipitation and works under the assumption that there is a monotone increase of PhiDP. In my opinion this algorithm is not adequate to compute KDP in the solid precipitation. Negative KDP can be linked to important phenomena such as electrification.

5. It is not clear to me what the authors do if the values of the polarimetric variables fall out of the range provided for the normalization. It is also not clear to me how gaps in the data are treated.

---

## Referee Comment (RC2) · Anonymous Referee #2 · 19 Sep 2019

Review of "Unsupervised classification of vertical profiles of dual polarization radar variables" by J. Tiira and D. N. Moisseev

Summary:

This manuscript details a method to classify vertical profiles of polarimetric radar observations in Finland over a three-year period. The classification method involves first processing the profiles by limiting the data to that above the melting layer, if one exists. The processed radar profiles are then transformed into principle components and subsequently clustered; the surface temperature is additionally included in the clustering for profiles without a melting layer. Based on the clustering results, the authors find

several distinct ice particle growth processes, including predominately pristine crystal growth, dendritic growth and subsequent aggregation, and Hallett-Mossop secondary ice production. Statistics of the clusters over a number of events are then presented.

Major comments:

The first major comment I have relates to producing separate classification methods for profiles with rain at the surface and profiles with snow at the surface. If the goal of this study is to identify ice microphysical processes from radar observations, it is unclear why similar processes occurring above the melting layer (for stratiform precipitation) should be identified separately from those some processes occurring in precipitation that happens to be snow at the surface. The authors have not demonstrated that ice growth in situations with rain at the surface is any different than ice growth with snow at the surface, other than the potential for increased aggregation just above the melting layer.

The second major comment I have is that the echo-top or cloud-top temperature of the profiles is more important to the ice growth processes (and is relevant in systems with either rain or snow a the surface) than the surface temperature. In fact, the presence of an inversion (found by the authors to be a common feature in their observed cases) could bias the clustering since the growth processes at upper levels of the atmosphere above the inversion have little relation to what is going on at the low levels. Having clusters essentially trained with a climatological lapse rate could then mistakenly assign profiles into the wrong growth regime if there is a strong deviation from the climatological temperature profiles during certain types of events. Some discussion of this point is warranted.

Minor comments:

1. Line 70: Add a description of the spatial and temporal resolution of the GDAS data.

2. Line 75: Clarify which radar variables have their medians calculated in linear space

and why this is being done; add reference if necessary.

3. Line 79: Does the method of KDP estimation impact the results? Add some discussion here of why this method was chosen for this study.

Line 100: How robust is the melting layer detection algorithm used by the authors if it requires the 4200-m threshold to limit detected peaks above this altitude. Some further discussion is needed here.

Line 104: Unsure what the sentence means, please clarify.

Lines 113-120: How much information does the surface temperature contain about the ice growth processes aloft? The authors should demonstrate that this surface temperature is a necessary component of the classification algorithm that improves its performance. A comparison between the clustering with the surface temperature and without the surface temperature, or similar test, would be informative.

Lines 133-135: Discuss how the PCA is performed for profiles with different numbers of bins. For example, when truncated data above the melting layer, the melting layer top will have different heights for different cases, resulting in profiles with non-uniform bins.

Lines 138-140: How were these standardization ranges chosen? Some of the upper bounds on these ranges such as for KDP and ZDR seem like they could be exceeded for C-band radar observations in certain conditions. Please discuss this further.

Line 148: Show a plot of the first principle components to provide a physical intuition of what these components represent. Also, the need for 30 principle components implies that adequately reducing the dimensionality of the radar observations is difficult. Explain how PCA is better than simply sampling the radar variables at various heights.

Lines 187-188: The description of the cluster convergence test here is unclear. Please clarify.

[Figure]

Lines 198-199: Are different clusters with similar profile shapes but with different magnitudes unique fingerprints of microphysical processes?

Lines 210-211: Do the authors have a specific application in mind when choosing the number of clusters to use in their study? Please discuss.

Line 216: "The order of components..." rather than "The component"?

Line 221: It may be more accurate to call this relation between peak ZDR and KDP and offset rather than an anticorrelation.

Figure 2: For cluster R3, the ZDR seems quite high so close to the melting layer. Please add some further discussion about this signature.

Line 234: Refer to figures in order; here, figures 7 and 8 are referred to before figures 5 and 6.

Line 235: Add some discussion here about whether polarimetric signatures of convection such as KDP and ZDR columns may be present in the radar observations, and therefore reflected in the classes.

Figure 3: Some of the classes are quite similar to each other (e.g., S9 and S10) to the point that it would be difficult to argue that they represent any unique fingerprints and instead reflect natural variability of the same microphysical process.

Lines 242-243: KDP values of 0.02 degrees/km and 0.04 degrees/km are both relatively low; please discuss further how this variable meaningfully separate the R0-3 and R4-5 classes. Also, what does the subscript "c" indicate with respect to the radar variables (found here and later instances)?

Figure 5: Explain why the ZDR centroid curve doesn't correspond to the profiles (i.e., shaded region) above 6 km.

Line 245: How common is it for precipitation systems in this region to have moist adiabatic lapse rates?

Lines 251-252: How do the authors separate out the contributions to KDP of planar crystals generated aloft and sedimenting from the KDP produced by secondary ice from the Hallett-Mossop process?

Lines 272-274: The maximum KDP values for the S14 class are much smaller than those for the S15 class, and the heights of the maximum ZDR values between these classes are also different. Discuss how these different profiles can both indicate a similar fingerprint of dendritic growth.

Line 283: Clarify here whether both processes are occurring within the same profile but at different heights or there are dendrites that have fallen into the Hallett-Mossop region where that processs is also ongoing.

Line 300: Add the color scale indicating the classes to figures 7 and 8.

Figure 7: Explanation for negative ZDR just above melting layer?

Figure 8: The ZDR profiles for this case appear fairly noisy. It might be helpful to show an RHI for this case, especially from the early portion of the event, to understand how much of the variability in the ZDR profiles is due to noise in the actual radar data.

Line 371: If these archetypes represent the desired output of the classification algorithm, doesn't this imply that the number of clusters (10 and 16) used in the classification algorithm is too high? Why not use 7 clusters in the algorithm?

Line 389: It would be beneficial to add some discussion here of the potential to identify the process of heavy riming, where ZDR values may become negative, as well as some indication of how common this process is in Southern Finland.

―――――――――――――――――――――――

---

## Author Comment (AC1) · 12 Nov 2019

**Revision and response to Anonymous Referee 1**

**Comments by Referee 1 and authors' response**

1. Generally speaking the paper is well written but it is clear that the authors do not have English as their mother language. In particular, oftentimes articles are missing in the sentence. This does not hinder the understanding of the text so I will not go through the list of what I have spotted but a conscious correction should be performed prior to publication

In the new iteration of the manuscript, we made several language corrections with special attention on the articles. With our previous papers in Copernicus publications, we've had a good experience with the last language check phase in helping with the final language corrections.

2. My main concern is with the method used for the clustering. To my understanding, the k-means algorithm expect all clusters to be of similar size. This is an unreasonable assumption in the case of weather phenomena since there are processes that are fairly common whereas others happen rarely. The authors, rightly, do not make any a-priori attempt to balance the data but I suspect that leads to classes that are a mix of various phenomena and hence difficult to link to specific microphysical processes. Other clustering methods such as Expectation Maximization (EM) clustering may be more adequate for the sort of data at hand.

The definition of cluster size is important here. By cluster size, the Referee 1 may refer to at least two different cluster properties: either the d-dimensional *area* occupied by the cluster, where d is the number of features (in our case, the number of principal components, d=30), or the *cardinality*. In k-means clustering, neighbouring clusters will occupy similar areas. However, the density of the points in the PCA space, and thus the cardinalities, may be different between the classes. Such is the case with the classification models presented in this manuscript; e.g. class S9 has 619 members while class S15 has only 16.

Using a clustering method that tends to produce classes occupying similar areas in the PCA space indeed involves a risk of suboptimal separation of the microphysical processes. In the study, we make an effort to address this problem by allowing a rather large number of initial classes and proposing that similar classes may be combined by identifying archetypes based on known fingerprints of the processes.

In the two classification models there are classes which may represent a mix of dendritic growth and secondary ice production. On the other hand, further examination of such profiles revealed that KDP signatures appearing both in the DGL and in the H-M region within the same profile is not uncommon, as described in the manuscript.

We added further discussion on these considerations in the revised manuscript.

3. Given that microphysical processes in the solid phase of precipitation are highly dependent on the ambient temperature and the authors have available an estimate of the temperature profile via the NCEP GDAS I would be very interested in having a look at the results of the clustering when including the full temperature profile instead of just the surface temperature.

There are various interesting and potentially useful ways in which the classification method introduced in this manuscript may be modified. However, the aim of this study is not to make a comprehensive comparison of several promising methods, but rather, presenting a reasoned but simple method as a starting point in studying vertical profiles through unsupervised classification and for its future applications.

In the revised manuscript we mention the use of alternative clustering methods and full temperature profiles as potential ways of further development of the method.

4. I am a bit surprised by the choice of algorithm to compute KDP. The Maesaka algo- rithm targets primarily the liquid layer of precipitation and works under the assumption that there is a monotone increase of PhiDP. In my opinion this algorithm is not adequate to compute KDP in the solid precipitation. Negative KDP can be linked to important phenomena such as electrification.

The Maesaka algorithm should be avoided when studying phenomena linked to negative KDP, which, however, is not the case in this study. In the present study, the main focus is on identification of processes typically occuring in stratiform precipitation, such as dendritic growth and the H-M process, for which negative KDP is not relevant, as explained in the revised manuscript.

5. It is not clear to me what the authors do if the values of the polarimetric variables fall out of the range provided for the normalization. It is also not clear to me how gaps in the data are treated.

The normalized values are not capped to 1, which is now explained in the text. There were no gaps in the measurements during the studied precipitation events. Since the classification is unaware of the temporal evolution of the profiles, a missing profile would not affect the ability to classify any other profile.

[revised manuscript text omitted]

---

## Author Comment (AC2) · 12 Nov 2019

Dear editors and Referee #2,

The authors would like to thank the editor and the reviewers for comprehensive comments and the opportunity to revise our manuscript 'Unsupervised classification of vertical profiles of dual polarization radar variables' (amt-2019-307). We are exceedingly grateful for the suggestions from the reviewers. The comments brought our attention to different ways of improving the overall readability of the paper and to parts that needed further explanation.

I have included the reviewer comments in a supplemented PDF document and responded to them individually. The individual comments and responses are followed by the revised text, with changes highlighted with colours: deletions in red and additions in blue. Colorbars have been added to Figures 7 and 8, as requested by Referee #2. The response to the individual comments includes two requested figures. As the manuscript itself already contains 9 figures, we feel that these additional visualizations are better left in this public response for the sake of concision of the paper.

We would like to express our great appreciation to the reviewers for the comments on our manuscript. We hope that the changes we have made resolve all your concerns about the article. We are happy to make any further changes that will improve the paper or facilitate successful publication.

Best regards,

Jussi Tiira

Please also note the supplement to this comment:
https://www.atmos-meas-tech-discuss.net/amt-2019-307/amt-2019-307-AC2-supplement.pdf

**Supplement:**

**Revision and response to Anonymous Referee 2**

**Major comments**

The first major comment I have relates to producing separate classification methods for profiles with rain at the surface and profiles with snow at the surface. If the goal of this study is to identify ice microphysical processes from radar observations, it is unclear why similar processes occurring above the melting layer (for stratiform precipitation) should be identified separately from those some processes occurring in precipitation that happens to be snow at the surface. The authors have not demonstrated that ice growth in situations with rain at the surface is any different than ice growth with snow at the surface, other than the potential for increased aggregation just above the melting layer.

By having separate classification based on surface precipitation type we do not intend to suggest that there is a fundamental difference in ice growth linked to the surface precipitation type. The reasons for separate processing are more technical. The most important reason is the difference in preprocessing. Because of the conditional truncation based on the occurrence of ML, there is a major difference in what the height represents in the profiles: either it is true altitude from the surface (S-model) or the distance from the ML (R-model). We are hesitant to mix the truncated and non-truncated profiles in the clustering phase fearing that the differences rising from the different preprocessing could drive the clustering.

Having separate methods brings some added complexity to the methodology. However, it does not prevent a holistic, all season analysis of the ice processes. Similar classes within and between the two classification models can be grouped as suggested in Section 5, where we introduce the class archetypes.

We didn't expect major seasonal differences in ice process fingerprints. However, we demonstrate a clear seasonal difference in KDP.

The second major comment I have is that the echo-top or cloud-top temperature of the profiles is more important to the ice growth processes (and is relevant in systems with either rain or snow a the surface) than the surface temperature. In fact, the pres- ence of an inversion (found by the

> authors to be a common feature in their observed cases) could bias the
> clustering since the growth processes at upper levels of the at- mosphere
> above the inversion have little relation to what is going on at the low levels.
> Having clusters essentially trained with a climatological lapse rate could
> then mistak- enly assign profiles into the wrong growth regime if there is a
> strong deviation from the climatological temperature profiles during certain
> types of events. Some discussion of this point is warranted.

The goal of this study is to see if unsupervised classificaiton can be used to document the characteristic profiles corresponding to snow growth processes. The answer to this seems to be yes, since we were able to identify such previously undocumented features as snow processes in inversion layers. Using the cloud top temperature, on the other hand, would not allow to diagnose such phenomena in the inversion layers.

It would be possible to use the whole temperature profiles as a classification feature, which might allow more accurate separation of the processes to different classes. This should be investigated in a future study.

We added discussion on this in Section 5.

**Minor comments**

> Line 70: Add a description of the spatial and temporal resolution of the
> GDAS data.

Added.

> Line 75: Clarify which radar variables have their medians calculated in
> linear space and why this is being done; add reference if necessary.

The mention of "linear space" here was a deprecated remnant from an earlier draft of the manuscript, where we had used means instead of medians. This mention is now removed, as a medians taken in dB and linear space are equal.

> Line 79: Does the method of KDP estimation impact the results? Add
> some discus- sion here of why this method was chosen for this study.

Essentially, the drawback of this method is the inability to produce negative KDP. However, negative KDP is not relevant for this study, as now explained in the revised text.

> Line 100: How robust is the melting layer detection algorithm used by the
> authors if it requires the 4200-m threshold to limit detected peaks above
> this altitude. Some further discussion is needed here.

We added a paragraph discussing the motivation for the thresholds used in steps 2. and 3.

Line 104: Unsure what the sentence means, please clarify.

This sentence has been reworded. Further, we added the definition of peak prominence and a reference to the implementation of the peak detection algorithm used.

Lines 113-120: How much information does the surface temperature contain about the ice growth processes aloft? The authors should demonstrate that this surface temperature is a necessary component of the classification algorithm that improves its performance. A comparison between the clustering with the surface temperature and without the surface temperature, or similar test, would be informative.

With the use of surface temperature as a classification feature, we show that KDP signatures occur near the surface in strong inversion layers in temperatures favored by dendritic growth as manifested in class S13. The use of surface temperature is necessary to distinguish this from the fingerprints of the H-M process. This is now discussed in better detail in Section 5.

Lines 133-135: Discuss how the PCA is performed for profiles with different numbers of bins. For example, when truncated data above the melting layer, the melting layer top will have different heights for different cases, resulting in profiles with non-uniform bins.

The numbers of bins are always 588 and 582 for profiles in snow and rain events, respectively. We made edits to Sections 2.1, 2.2.1 and 3.1 to better explain this. Please note that for rain events, the highest bin is 10 km above the ML top rather than at 10 km altitude. So, precisely speaking, we are rather shifting the observation windows of the profiles than truncating them. We feel, however, that "truncation" is an appropriate term to describe what this effectively does, as echo tops are usually lower than the ML top height + 10 km.

Lines 138-140: How were these standardization ranges chosen? Some of the upper bounds on these ranges such as for KDP and ZDR seem like they could be exceeded for C-band radar observations in certain conditions. Please discuss this further.

Added a sentence here: "The values of the standardized variables are not capped, but values greater than 1 are allowed when the unscaled values exceed b."

Line 148: Show a plot of the first principle components to provide a physical intuition of what these components represent. Also, the need for 30 principle components implies that adequately reducing the dimensionality of the radar observations is difficult. Explain how PCA is better than simply sampling the radar variables at various heights.

There are several incentives for using PCA over sampling some of which are now discussed in the revised manuscript. We expect that the total number of samples needed would be even higher than 30 (=10 per radar variable). PCA

also optimizes the weights of the components for clustering, since the variance of each component reflects how much variance they explain in the original data.

The first components are show in Figure 1 below.

[Figure]

Figure 1: The first PCA components of R- and S-models

> Lines 187-188: The description of the cluster convergence test here is unclear. Please clarify.

We made a small clarification here. The convergence of the clustering solutions can be analyzed by simply comparing the cluster centroids of each repetition of the clustering process.

> Lines 198-199: Are different clusters with similar profile shapes but with different mag- nitudes unique fingerprints of microphysical processes?

We don't consider them unique fingerprints. This is why they are here termed trivial differences. We added a sentence in this paragraph to highlight the goal of evaluating the differences.

> Lines 210-211: Do the authors have a specific application in mind when choosing the number of clusters to use in their study? Please discuss.

Good point. We now mention here our objective of separating the main profile characteristics and the process fingerprints to individual classes.

> Line 216: "The order of components. . . " rather than "The component"?

Here we are referring to the value of the first principal component. In PCA, the principal components are by definition ordered in the decreasing order of their variance (and the explained variability in the original data). We made some rephrasing in this paragraph to avoid confusion between the concepts of a principal component and a profile class.

> Line 221: It may be more accurate to call this relation between peak ZDR and KDP and offset rather than an anticorrelation.

In this paragraph we are discussing the high values of ZDR and KDP (and their association to echo top height) rather than the altitudes where those values occur. The beginning of the paragraph is rephrased to avoid confusion.

> Figure 2: For cluster R3, the ZDR seems quite high so close to the melting layer. Please add some further discussion about this signature.

Discussion on this feature can be found in the the paragraph starting from the line 253 in the first revision of the discussion paper. We made small edits to the paragraph to point the reader to Fig. 3 to highlight the finding that this signature is stronly linked to the echo top being in the DGL.

> Line 234: Refer to figures in order; here, figures 7 and 8 are referred to before figures 5 and 6.

Added a note that Figs. 7 and 8 are introduced later. Here we simply note the reader that two different KDP ranges are used in the figures.

> Line 235: Add some discussion here about whether polarimetric signatures of convec- tion such as KDP and ZDR columns may be present in the radar observations, and therefore reflected in the classes.

Added.

> Figure 3: Some of the classes are quite similar to each other (e.g., S9 and S10) to the point that it would be difficult to argue that they represent any unique fingerprints and instead reflect natural variability of the same microphysical process.

Some classes are indeed quite similar. With surface temperature as an additional clustering feature this is difficult to avoid. The mentioned S9 and S10 highlight this characteristic of the method, since the only significant difference between them seems to be the surface temperature. In the initial revision, we discuss similarities and combining classes in section 5, but indeed we should mention this consideration already much earlier in the paper. We added a paragraph about this in the end of Section 3.3.

> Lines 242-243: KDP values of 0.02 degrees/km and 0.04 degrees/km are both relatively low; please discuss further how this variable meaningfully separate the R0-3 and R4-5 classes. Also, what does the subscript "c" indicate with respect to the radar variables (found here and later instances)?

In this paragraph we discuss the KDP related characteristics of each class. However, as pointed out in Sections 2 and 3, the classification is based on not only profiles of KDP, but also ZDR, and Z. Hence, the classes are not required to have unique characteristics in KDP, but rather in the combination of the three

variables. These other characteristics are also discussed in this section. We have rewritten this paragraph to improve its readability, explaining also the subscript "c", which is used for referring to cluster centroid values instead of the measured values in individual profiles.

> Figure 5: Explain why the ZDR centroid curve doesn't correspond to the profiles (i.e., shaded region) above 6 km.

By following the shaded areas, we see that at least 75 % of the profiles in class S13 have echo tops lower than 7 km. However, there are some profiles with higher echo tops. Many of those profiles have considerable values of ZDR above the 7 km altitude, and thus the class centroid has values above zero. We would see a similar mismatch if we would make a comparison between the quantiles and the mean.

> Line 245: How common is it for precipitation systems in this region to have moist adiabatic lapse rates?

Moist adiabatic lapse rate was used here as the first approximation to give the reader a rough idea of the ambient temperatures around the peak KDP height of the R6 and R9 profiles. For a better estimate, and to be consistent with the rest of the manuscript, we now give the class mean of NCEP GDAS value at the peak centroid KDP altitude in the rewritten paragraph.

> Lines 251-252: How do the authors separate out the contributions to KDP of planar crystals generated aloft and sedimenting from the KDP produced by secondary ice from the Hallett-Mossop process?

For a single profile, it is difficult to do such separation. Class centroids, on the other hand, represent statistical features of the member profiles. A slighly bimodal centroid KDP with peaks corresponding to DGL and H-M, for example, is a very likely indicator of the contribution of the H-M process. This is now mentioned a couple of paragraphs down where classes R7 and S12 are discussed.

> Lines 272-274: The maximum KDP values for the S14 class are much smaller than those for the S15 class, and the heights of the maximum ZDR values between these classes are also different. Discuss how these different profiles can both indicate a similar fingerprint of dendritic growth.

We added a couple of sentences here discussing the differences and the likely reasons behind them.

> Line 283: Clarify here whether both processes are occurring within the same profile but at different heights or there are dendrites that have fallen into the Hallett-Mossop region where that processs is also ongoing.

This is now clarified as mentioned in an answer above.

> Line 300: Add the color scale indicating the classes to figures 7 and 8.

Added.

Figure 7: Explanation for negative ZDR just above melting layer?

Unfortunately we have no explanation for this. We frequently observe a similar dip in ZDR just above the melting layer, but we have not studied this further. There is a possibility of a small calibration bias which could explain why the lowest values are slightly negative instead of closer to zero.

Figure 8: The ZDR profiles for this case appear fairly noisy. It might be helpful to show an RHI for this case, especially from the early portion of the event, to understand how much of the variability in the ZDR profiles is due to noise in the actual radar data.

The RHI of ZDR, as shown in the attached Figure 2, has heavy noise in the upper part. The ZDR indeed still looks rather noisy even after total averaging range of 2 km used in this study.

[Figure]

Figure 2: RHI of ZDR. Hyytiälä measurement station is at 64 km.

Line 371: If these archetypes represent the desired output of the classification algorithm, doesn't this imply that the number of clusters (10 and 16) used in the classification algorithm is too high? Why not use 7 clusters in the algorithm?

This is now discussed in Section 5 in the revised manuscript.

> Line 389: It would be beneficial to add some discussion here of the potential to identify the process of heavy riming, where ZDR values may become negative, as well as some indication of how common this process is in Southern Finland.

Added.

**Other corrections**

On lines 335-336 of the first version of the discussion paper, the numbers of events manually classified as either stratiform or convective were wrong. These numbers reflected a different way of separating individual events which was used in an earlier draft of the manuscript. The numbers have now been corrected to reflect the current definition of individual events stated on lines 240-241 of the first discussion version.

[revised manuscript text omitted]

---

## Referee Report (RR1)

**Review of "Unsupervised classification of vertical profiles of dual polarization radar variables" by J. Tiira and D. N. Moisseev**

**Summary:**

This manuscript is generally improved from the previous version. However, I still have some general and specific comments to address at various sections of the manuscript. These comments should be addressed before this manuscript is published.

**General comments:**

Add some discussion about the impact of vertical wind shear on the variability of radar signatures of the same physical processes. The differential advection of hydrometeors at different height levels will produce fall streaks that are only partially sampled by a vertical profile (or an RHI for that matter). As such, gradients in the vertical profiles of the radar variables may be strongly impacted by the wind profile and not reflect purely microphysical processes. I think some caution is therefore warranted in interpreting the microphysical processes corresponding to the centroid profiles.

Specific comments (line numbers here correspond to the updated version of the manuscript with the changes shown):

1. Line 46: More accurate to say that high ZDR is linked to the presence of planar crystals, where the presence of supercooled liquid at these temperatures indicates very favorable conditions for their growth.

2. Lines 48-50 (and elsewhere throughout the manuscript): How can horizontal banded features be determined from a vertical profile? Please clarify.

3. Line 85: Negative KDP has also been documented during periods of conical graupel (see Oue et al. 2015). Also add citation for negative KDP from ice particles aligned by an ambient electric field.

4. Line 88: Please clarify whether the noise and clutter filtering is done on the RHI data or on the binned profiles.

5. Line 105: Add the name of the peak detection function from the SciPy package in a footnote.

6. Line 119: Would these thresholds eliminate dendritic growth zone "bright bands" from being erroneously labeled as melting layers? Aren't some of the dendritic growth zones revealed by the analysis at heights below 4.6 km?

7. Line 123: Discuss whether the smoothing in time corresponds to a specific temporal scale.

8. Line 139: The phrase "10-km layer from the lowest elevation of 200 m" refers to 0.2-10.0 km, correct? If so, this sentence should be changed to "...layer between 0.2 km and 10 km." Also indicate whether heights are with respect to the ground or mean sea level.

9. Line 177: Shouldn't the standard deviations here refer to P(Ts) not Ts?

10. Line 190: Remove "but non-trivial."

11. Lines 214-215: I am unsure what this sentence means. Please rephrase so that the reader can understand how the centroid profiles change by adding additional clusters.

12. Lines 230-231: Please add some examples of how the optimal number of classes changes for specific applications. This could be added here or in the conclusions section of the paper where this phrase is also used.

13. 267-268: The seasonal variability of vertical motion with temperature level could also impact the magnitude of the ZDR and KDP enhancements.

14. Line 269-271: Is there a citation the authors can reference to discuss the seasonal climatology of convective precipitation in Finland?

15. Line 279: Rephrase to "With respect to KDP intensity..."

16. Line 316: Add that heights of maximum ZDR between S14 and S15 are different.

17. Figure 8: Remove mention of the melting layer height in the caption for this all-snow case.

18. Figure 9: For panel a2, please label or explain the units. Shouldn't relative frequencies be less than 1?

19. Line 324: Add the microphysics parameterization used in the Sinclair et al. (2016) study.

20. Line 390: Change "also represent considerable" to "represent more modest."

21. Line 402: What is meant by "data-driven?" Please elaborate.

22. Line 467: Add some discussion of whether riming is represented in any of the classes produced by the clustering algorithm. If this process occurs with relatively frequency in the region of the observations, it should either be represented in some of the classes or simply not distinguishable from aggregation in unsupervised classification method.

**References:**

Oue, M., M. R. Kumjian, Y. Lu, Z. Jiang, E. E. Clothiaux, J. Verlinde, and K. Aydin, 2015: X-band polarimetric and Ka-band Doppler spectral radar observations of a graupel-producing Arctic mixed-phase cloud. *J. Appl. Meteor. Climatol*, **54**, 1335–1351, doi:https://doi.org/10.1175/JAMC-D-14-0315.1.

---

## Author Response (AR2)

**Revision and response to reviewers**

Dear editor,

The authors wish to thank the reviewers for their time and effort in giving these additional suggestions for further improving our manuscript 'Unsupervised classification of vertical profiles of dual polarization radar variables'. We found the comments very useful and have made changes accordingly.

Similarly to the previous submission, we have included the reviewers' comments in this document and responded to them individually. The individual comments and responses are followed by the revised text, with changes highlighted with colours: deletions in red and additions in blue.

We hope that the changes we have made resolve all your concerns about the article. We are happy to make any further changes that will improve the paper or facilitate successful publication.

Best regards,

Jussi Tiira

**Comments by Referee 1 and author's response**

> I am satisfied on how the authors have addressed points 1 and 5 of my previous review. I would still be terribly interested in seeing the results of the classification when using a full temperature profile as I mention in point 3, although I understand that this is an exploratory work and sub-sequent papers can further expand the methodology.

We agree and we would be interested in conducting such study as well.

> I am reasonably satisfied of the explanation given on the choice of the k-means algorithm (point 2) in the response to the authors. However, I would like this explanation to be more explicitly mentioned in the conclusion of the paper instead of simply brushing it aside by saying that other methods could be explored. There are enough reasons to think that some classes are a mix of very different processes that could be separated using other techniques.

A good suggestion. We added the key points of the explanation in the discussion.

> I think the choice of KDP algorithm was simply a mistake by the authors since this algorithm was designed explicitly for rain. Ideally, the authors should repeat the data processing by using another more generic estimator such as the least square method since there is no way to know a-priori the impact of the choice of algorithm. Could the authors at least present a scatter plot of the Maesaka KDP versus another more adequate KDP algorithm to show whether there are significant differences between the KDP obtained? Actually, this is part of the broader topic of the sensitivity of the results to the data quality, which has not been covered by the paper. If the algorithm is over-sensitive to biases and noise its use may be limited in practical applications. From my point of view, the paper can be published if the authors address the KDP issue properly.

When initially choosing a KDP algorithm for this study, we tried a couple of alternatives, namely the implementations `pyart.retrieve.kdp_schneebeli` and `pyart.retrieve.kdp_vulpiani` in the Py-Art package (Helmus and Collis, 2016). These alternatives produced more noisy KDP, and the final result could have been quite sensitive to the heavy preprocessing methods needed for reproducible results. However, the values underlying the noise seemed comparable with the smoother results produced using the Maesaka algorithm, which is why we were confident choosing it despite it's limitations.

Following your suggestion, we had another go on finding a combination of PHIDP preprocessing and a KDP algorithm that would produce a stable estimate for KDP, while allowing negative values. A comparison between the Maesaka KDP and an alternative using a method developed in the CSU Department of Electrical Engineering (Lang et.al., 2019) is shown in Figures 1 and 2. In Figure 1 visualizing the 21 February 2014 case, the only major difference are the negative kdp values around 20 UTC, which the Maesaka algorithm is unable to produce. The scatterplot in Figure 2 highlights the good agreement between the results from the two KDP methods.

**References**

Timothy Lang, Brenda Dolan, Nick Guy, CAM Gerlach and Joseph Hardin: `CSU-Radarmet/CSU_RadarTools: CSU_RadarTools v1.3`, doi:10.5281/zenodo.2562063, 2019.

[Figure]

Figure 1: Timeseries of vertical profiles of KDP calculated using the Maesaka and CSU algorithms in the top and middle panels, respectively, and reflectivity in the bottom panel.

[Figure]

Figure 2: A scatterplot to comparing the Maesaka KDP on the x-axis and the CSU KDP on the y-axis for the 21 February 2014 case, and a 1:1 reference line in black.

**Comments by Referee 2 and authors' response**

**General comments**

> Add some discussion about the impact of vertical wind shear on the variability of radar signatures of the same physical processes. The differential advection of hydrometeors at different height levels will produce fall streaks that are only partially sampled by a vertical profile (or an RHI for that matter). As such, gradients in the vertical profiles of the radar variables may be strongly impacted by the wind profile and not reflect purely microphysical processes. I think some caution is therefore warranted in interpreting the microphysical processes corresponding to the centroid profiles.

We agree. Adjustments for wind shear become more viable if volume scans are utilized. This is something we have started working on as a follow up study.

Added a paragraph in the discussion:

"It should be noted that wind shear effects induce differential advection of hydrometeors at different altitudes affecting the gradients in the vertical profiles of radar variables (Lauri et.al., 2012). Therefore caution should be used in interpreting microphysical processes corresponding to class centroid profiles. The wind shear effects are difficult to correct for using vertical profile or RHI radar observations due to the limitations in horizontal sampling. Such adjustments become more viable if classification is performed on profiles extracted from volume scans, which will be investigated in future work."

**Specific comments**

> 1. Line 46: More accurate to say that high ZDR is linked to the presence of planar crystals, where the presence of supercooled liquid at these temperatures indicates very favorable conditions for their growth.

We agree and edited the sentence according to your suggestion.

> 2. Lines 48-50 (and elsewhere throughout the manuscript): How can horizontal banded features be determined from a vertical profile? Please clarify.

Wide banded or layered structures are very typical for stratiform clouds. Almost all winter precipitation events in Southern Finland, including the ones studied in this paper, originate from stratiform clouds. Therefore, in the context of winter events, it is safe to assume that peaks in vertical profiles of radar observations correspond to such banded features.

You are, however, correct to point out that in many occasions, we seemed to have used the term "band" almost as a synonyme of a "peak" or a "layer".

Througout the manuscript, we made changes in the use of such terminology to avoid confusion.

> 3. Line 85: Negative KDP has also been documented during periods of conical graupel (see Oue et al. 2015). Also add citation for negative KDP from ice particles aligned by an ambient electric field.

Thank you for pointing this out. This is important to keep in mind in any future studies linking profile classes to riming. We added the suggested citations.

Both conical graupel and electic field induced vertical alignment events are short lived. Therefore they have limited impact in the long term statistics, as pointed out in the revised text.

> 4. Line 88: Please clarify whether the noise and clutter filtering is done on the RHI data or on the binned profiles.

It now says "applied to the binned profiles".

> 5. Line 105: Add the name of the peak detection function from the SciPy package in a footnote.

Added.

> 6. Line 119: Would these thresholds eliminate dendritic growth zone "bright bands" from being erroneously labeled as melting layers? Aren't some of the dendritic growth zones revealed by the analysis at heights below 4.6 km?

The use of these thresholds is an additional automated quality control step that was found to reduce errors in ML detection. Such dendritig growth layer "bright bands" are usually weaker than the actual melting layer signal, at least in majority of the profiles during a case. The threshold that prevents dramatic changes in ML height will therefore eliminate most of the DGL "bright bands" from being labeled as ML. The 4.6 km threshold was found to eliminate few additional errors, but plays a less important role here. However, as any of the other ML detection algorithms out there, our method is not perfect, and there indeed were a couple of rain events where a DGL "bright bands" were labeled as ML. For this study, we manually analyzed all the results of the ML detection and discarded any events where ML was not correctly detected. This seems to be only briefly mentioned in the beginning of second paragraph of Section 2 in the last reviewed revision. Overall, we are very happy with the accurracy of our ML detection method; In over 90% of the events in the original data set, no errors in the ML detection were found.

Added "The results from the ML detection were analyzed manually and the events with errors were discarded. In 90% of events in the original data set, ML was detected without errors."

> 7. Line 123: Discuss whether the smoothing in time corresponds to a specific temporal scale.

Edited to: "A rolling triangle mean with a window size of 5 profiles, correspoding to one hour, is used for smoothing."

> 8. Line 139: The phrase "10-km layer from the lowest elevation of 200 m" refers to 0.2-10.0 km, correct? If so, this sentence should be changed to "…layer between 0.2 km and 10 km." Also indicate whether heights are with respect to the ground or mean sea level.

Edited accordingly. Heights are with respect to ground level.

> 9. Line 177: Shouldn't the standard deviations here refer to P(Ts) not Ts?

The target deviation on Ts is defined as median(2 std(Ts)) = 3 degC, the median being over the clusters. It is intended to be defined with respect to Ts and not P(Ts). Otherwise, the next sentence stating that, typically, 95% of Ts values in a cluster would be within 3 degC range from the cluster mean, would not hold true.

> 10. Line 190: Remove "but non-trivial."

Removed.

> 11. Lines 214-215: I am unsure what this sentence means. Please rephrase so that the reader can understand how the centroid profiles change by adding additional clusters.

Here we intend to explain that there are no dramatic changes in the cluster centroids if k is changed. Such sensitivity to k would be unexpected.

Rephrased to "Moreover, the properties of the cluster centroids are not highly sensitive to k. Clustering performed with k=k0 and k=k0 + 1 would typically result in sets of clusters sharing k0 - 1 to k0 very similar centroids."

> 12. Lines 230-231: Please add some examples of how the optimal number of classes changes for specific applications. This could be added here or in the conclusions section of the paper where this phrase is also used.

There is a possibility that the optimal number of clusters depends on the application, but we would like to point out that we have not tested it yet. This is a topic of our current research, where we are using the classification for adaptive selection of Z-S relations. We have observed that some of the classes have similar relations, therefore in such cases, the number of clusters can be reduced. This is work in progress. We don't know yet if we should reduce the number of clusters by reducing k or combining classes in this specific application.

13. 267-268: The seasonal variability of vertical motion with temperature level could also impact the magnitude of the ZDR and KDP enhancements.

Added a sentence mentioning this following your suggestion.

14. Line 269-271: Is there a citation the authors can reference to discuss the seasonal climatology of convective precipitation in Finland?

Two citations were added here referring to studies discussing dual-polarization weather radar singatures in convective systems (Voormansik et.al., 2017) and seasonal climatology of lightning and convection in Northern Europe (Mäkelä et.al., 2014).

15. Line 279: Rephrase to "With respect to KDP intensity..."

Rephrased.

16. Line 316: Add that heights of maximum ZDR between S14 and S15 are different.

Added according to your suggestion.

17. Figure 8: Remove mention of the melting layer height in the caption for this all-snow case.

Removed. Thank you for pointing this out.

18. Figure 9: For panel a2, please label or explain the units. Shouldn't relative frequencies be less than 1?

The panel a2 is relabeled and the units explained better in both the caption and in the main text. Essentially it shows the ratios of the number of profiles in convective cases per class to the expected value in uniform distribution, i.e. a value of 2 would mean that the given class has twice the amount of profiles in convective cases classified to it than you would expect from a uniform distribution.

19. Line 324: Add the microphysics parameterization used in the Sinclair et al. (2016) study.

They used the double-moment Morrison microphysics scheme (Morrison et al., 2005). Citation added.

20. Line 390: Change "also represent considerable" to "represent more modest."

Changed.

21. Line 402: What is meant by "data-driven?" Please elaborate.

I think we used the term here to highlight that in PCA is used as a data-driven method to extract classification features from the data.

"clustering of data-driven features" was changed to "clustering of PCA components".

> 22. Line 467: Add some discussion of whether riming is represented in any of the classes produced by the clustering algorithm. If this process occurs with relatively frequency in the region of the observations, it should either be represented in some of the classes or simply not distinguishable from aggregation in unsupervised classification method.

For the summer cases we have no reliable way of knowing wether riming occurs and to what extent, yet. For snowfall cases ground based measurements can be used for this. But because this paper combines analysis of summer and snow cases, we did not study how this classification could be used to study riming. Studying this would be a topic in itself, and because of the limitations in the lenght of the paper, this should be studied separately. Discussing how riming is represented in the classes without conducting such a study would be almost purely speculation, which we are hesitant to do.

[revised manuscript text omitted]